# Cortical microtubule pulling forces contribute to the union of the parental genomes in the *Caenorhabditis elegans* zygote

**Griselda Velez-Aguilera, Batool Ossareh-Nazari, Lucie Van Hove, Nicolas Joly, Lionel Pintard***

Université Paris cité, CNRS, Institut Jacques Monod, F-75013, Paris, France

**Abstract** Previously, we reported that the Polo-like kinase PLK-1 phosphorylates the single *Caenorhabditis elegans* lamin (LMN-1) to trigger lamina depolymerization during mitosis. We showed that this event is required to form a pronuclear envelope scission event that removes membranes on the juxtaposed oocyte and sperm pronuclear envelopes in the zygote, allowing the parental chromosomes to merge in a single nucleus after segregation (Velez-Aguilera et al., 2020). Here, we show that cortical microtubule pulling forces contribute to pronuclear envelopes scission by promoting mitotic spindle elongation, and conversely, nuclear envelopes remodeling facilitates spindle elongation. We also demonstrate that weakening the pronuclear envelopes via PLK-1-mediated lamina depolymerization, is a prerequisite for the astral microtubule pulling forces to trigger pronuclear membranes scission. Finally, we provide evidence that PLK-1 mainly acts via lamina depolymerization in this process. These observations thus indicate that temporal coordination between lamina depolymerization and mitotic spindle elongation facilitates pronuclear envelopes scission and parental genomes unification.

**\*For correspondence:**
Lionel.PINTARD@ijm.fr

**Competing interest:** The authors declare that no competing interests exist.

## Editor's evaluation

This manuscript shows how the mitotic spindle helps to break apart the nuclear envelopes surrounding the maternal and paternal genomes so that they can be mixed together after fertilisation. This study will be interesting for cell biologists and biophysicists studying nuclear organization and mechanics. The work provides new insights into how pulling forces from the cell cortex influence the dynamics of nuclear rupture during mitosis.

## Introduction

The life of sexually reproducing organisms starts with joining two haploid genomes. Parental chromosomes are first replicated in distinct pronuclei, each surrounded by a nuclear envelope, and meet for the first time during the first mitosis. Coordinated disassembly of the pronuclear envelopes is required to promote the reunification of the parental chromosomes in the fertilized zygote, but the underlying mechanisms are incompletely understood.

The *Caenorhabditis elegans* zygote provides an attractive model system to investigate the mechanisms by which the maternal and paternal genomes unify at the beginning of life (*Oegema and Hyman, 2006*; *Cohen-Fix and Askjaer, 2017*; *Pintard and Bowerman, 2019*). After fertilization, the oocyte and sperm chromosomes surrounded by a nuclear envelope localize to opposite sides of the zygote. The female pronucleus is located in the anterior, whereas the male pronucleus—and

associated centrosomes—is in the posterior. When the two pronuclei meet, they change their shape, flattening out along their juxtaposed sides, eventually aligning along the AP axis of the one-cell embryo (*Oegema and Hyman, 2006*). The two pronuclei undergo spatially regulated nuclear envelope breakdown (NEBD) to permit chromosome attachment to microtubules and their alignment on the metaphase plate. The mechanisms promoting NEBD and remodeling for the subsequent unification of two parental genomes in the first mitosis of the zygote remain poorly understood and are the focus of this study.

The nuclear envelope consists of two lipid bilayers: the inner nuclear membrane (INM) and outer nuclear membrane (ONM), separated by the perinuclear space (PNS) (*Cohen-Fix and Askjaer, 2017*). The ONM is continuous with the endoplasmic reticulum membrane, while the PNS is continuous with the ER lumen. Transport across the two nuclear membranes occurs through the nuclear pore complexes (NPCs), anchored to the membrane via several transmembrane nucleoporins. Underlying the INM, the nuclear lamina provides mechanical integrity to the nucleus. *C. elegans* encodes a single lamin (LMN-1), closer to vertebrate B-type than A-type lamins (*Liu et al., 2000*).

In the *C. elegans* zygote, mitosis is semi-open, and the nuclear envelope only partially breaks down during spindle assembly (*Cohen-Fix and Askjaer, 2017*). NPC disassembly and lamina depolymerization starts in the vicinity of the centrosomes and then later on the juxtaposed envelopes located between the parental chromosomes (*Lee et al., 2000*; *Hachet et al., 2012*; *Velez-Aguilera et al., 2020*). Complete removal of the NPCs and the lamina only occurs in early anaphase (*Lee et al., 2000*; *Hachet et al., 2012*; *Velez-Aguilera et al., 2020*). While NPCs disassemble and the lamina depolymerizes, remnant nuclear membranes, which contain INM and ONM proteins, persist between the parental chromosomes during mitosis. Removal of the membranes between the parental chromosomes begins with the formation of a membrane scission event (also called a membrane gap), visible by fluorescent microscopy by labeling INM proteins (e.g., LEM-2) (*Audhya et al., 2007*). This event occurs concomitantly with the parental chromosomes congressing on the metaphase plate, 10–40 s before anaphase onset (*Audhya et al., 2007*; *Rahman et al., 2020*), allowing the chromosomes from the two pronuclei to mingle on the metaphase plate and join in a single nucleus after chromosome segregation. How this membrane scission event forms remains poorly understood. Intriguingly, it always appears at an equal distance from the two centrosomes and depends on the proper alignment of the chromosomes on the metaphase plate (*Rahman et al., 2015*). Consistently, membrane scission is totally prevented in *hcp-3*<sup>CENPA</sup>(*RNAi*) embryos (*Rahman et al., 2015*) that fail to assemble kinetochores and are defective in chromosome alignment (*Oegema et al., 2001*).

Formation of this membrane gap also requires depolymerization of the lamina by the mitotic Polo-like kinase PLK-1 (*Rahman et al., 2015*; *Martino et al., 2017*; *Velez-Aguilera et al., 2020*). Accordingly, expression of an LMN-1 version, carrying eight non-phosphorylable alanines replacing serines (hereafter LMN-1 8A), is sufficient to prevent the formation of pronuclear envelopes scission. A failure to form this membranes scission can result in the appearance of embryos with a paired nuclei phenotype (*Rahman et al., 2015*; *Martino et al., 2017*; *Velez-Aguilera et al., 2020*). In these embryos, the two sets of parental chromosomes remain physically separated during mitosis and segregate into two separate DNA masses at each pole of the spindle. In turn, this leads to the formation of two nuclei in each cell of the two-cell embryo (paired nuclei) (*Audhya et al., 2007*; *Bahmanyar et al., 2014*; *Galy et al., 2008*; *Rahman et al., 2015*; *Martino et al., 2017*; *Velez-Aguilera et al., 2020*).

Besides PLK-1-mediated lamina depolymerization, mechanical forces provided by astral microtubules could also contribute to pronuclear envelopes scission by facilitating the removal of the lamina, similar to the situation in human cells (*Salina et al., 2002*; *Beaudouin et al., 2002*), but possibly also by promoting mitotic spindle elongation.

Here, we follow-up on our previous work showing that pronuclear envelopes scission is regulated via PLK-1-mediated lamina depolymerization (*Velez-Aguilera et al., 2020*) by testing whether cortical microtubule pulling forces also contribute to this process. We show that astral microtubule pulling forces contribute to pronuclear membranes scission by facilitating lamina disassembly and promoting mitotic spindle elongation. Our observations thus suggest that temporal coordination between chromosome alignment, lamina depolymerization, and spindle elongation induces pronuclear envelopes scission and the unification of the parental chromosomes after segregation.

## Results and discussion

### Microtubules dynamics during NEBD in the one-cell *C. elegans* embryo

Previous work established that NEBD is spatially regulated in the fertilized one-cell *C. elegans* zygote (*Lee et al., 2000*; *Hachet et al., 2012*; *Velez-Aguilera et al., 2020*), but the exact timing of each event and the contribution of microtubules to this process had not yet been investigated. To address this point, we simultaneously visualized microtubules and nuclear envelope dynamics with a high temporal resolution during NEBD. We used spinning disk confocal microscopy to film one-cell embryos (one image/2 s), expressing fluorescently labeled tubulin (GFP::TBB-2) with either the lamina (mCherry::LMN-1) (*Figure 1A*, *Figure 1—video 1*), or the transmembrane nucleoporin NPP-22$^{NDC1}$ (mCherry::NPP-22$^{NDC1}$) to visualize nuclear membranes (*Figure 1B*, *Figure 1—video 2*). We also filmed embryos expressing GFP::TBB-2 and mCherry-tagged histone (mCherry::Histone) to monitor the configuration of the chromosomes during the different steps of NEBD (*Figure 1C*, *Figure 1—video 3*).

Before NEBD, 200 s prior to anaphase onset, microtubules were excluded from the pronuclei space, and the pronuclear envelopes appeared flattened along their juxtaposed sides but presented elsewhere a rounded shape, except around centrosomes where the nuclear envelopes were highly curved (90° angle) (*Figure 1A and B*). Later, 160 s before anaphase onset, signs of nuclear envelope deformation were apparent in the vicinity of the centrosomes (*Figure 1A*, red arrow). At these sites, the pronuclear envelopes appeared deformed toward the centrosomes, with the appearance of protrusions, suggesting that microtubules, emanating from the centrosomes were pulling on the pronuclear envelopes. Pronuclear envelopes permeabilization, measured by the appearance of soluble tubulin in the pronuclei space, was detected around 160 s before anaphase onset, concomitantly with pronuclear membranes deformation near the centrosomes (*Figure 1A–D*, *Figure 1—source data 1*). Nuclear envelope permeabilization was systematically detected first in the male pronucleus (*Figure 1—figure supplement 1*). Soon after membrane permeabilization, the first microtubules started to invade the pronuclei space. As more microtubules invaded the pronuclei space to capture paternal chromosomes, the overall surface of the pronuclei decreased (*Figure 1E*, *Figure 1—source data 1*), and the pronuclei adopted a more triangular shape, possibly dictated by microtubules assembling the mitotic spindle in the space confined by the remnants of the pronuclear envelopes (*Figure 1B*; *Hayashi et al., 2012*).

Signs of lamina disappearance from the pronuclear envelopes systematically started in the vicinity of centrosomes, 70 s before anaphase onset (*Figure 1A and D*, *Figure 1—figure supplement 2*), possibly as a consequence of microtubule pulling at this site, together with the action of the lamin kinase PLK-1, which is enriched at centrosomes (*Chase et al., 2000*; *Budirahardja and Gönczy, 2008*; *Nishi et al., 2008*; *Martino et al., 2017*). The lamina progressively disappeared in the vicinity of the centrosomes but persisted between the parental chromosomes. Lamina disassembly between the pronuclei started only 50 s before anaphase onset (*Figure 1A and D*). At this time point, the lamina network was still detected on the nuclear envelope remnants surrounding the chromosomes (*Figure 1A*). The membrane scission event between the juxtaposed pronuclei was detected later, around 30 s before anaphase onset (*Figure 1B and D*, *Figure 1—figure supplement 3*), consistent with previous observations (*Audhya et al., 2007*). The membrane scission event appeared immediately after chromosome alignment on the metaphase plate (*Figure 1B*, red arrowhead, *Figure 1—figure supplement 3*, *Figure 1—source data 1*).

Starting 80 s before anaphase onset, the distance between the centrosomes steadily increased as a result of spindle elongation (*Figure 1F*, *Figure 1—source data 1*). The beginning of spindle elongation was concomitant with lamina depolymerization but preceded membrane gap formation. Based on these observations, we hypothesized that by pulling on nuclear envelopes and membranes, and by tearing apart the lamina and by elongating the mitotic spindle, astral microtubule pulling forces might contribute to pronuclear envelopes breakdown, and thus to the unification of the parental chromosomes in the early *C. elegans* embryo.

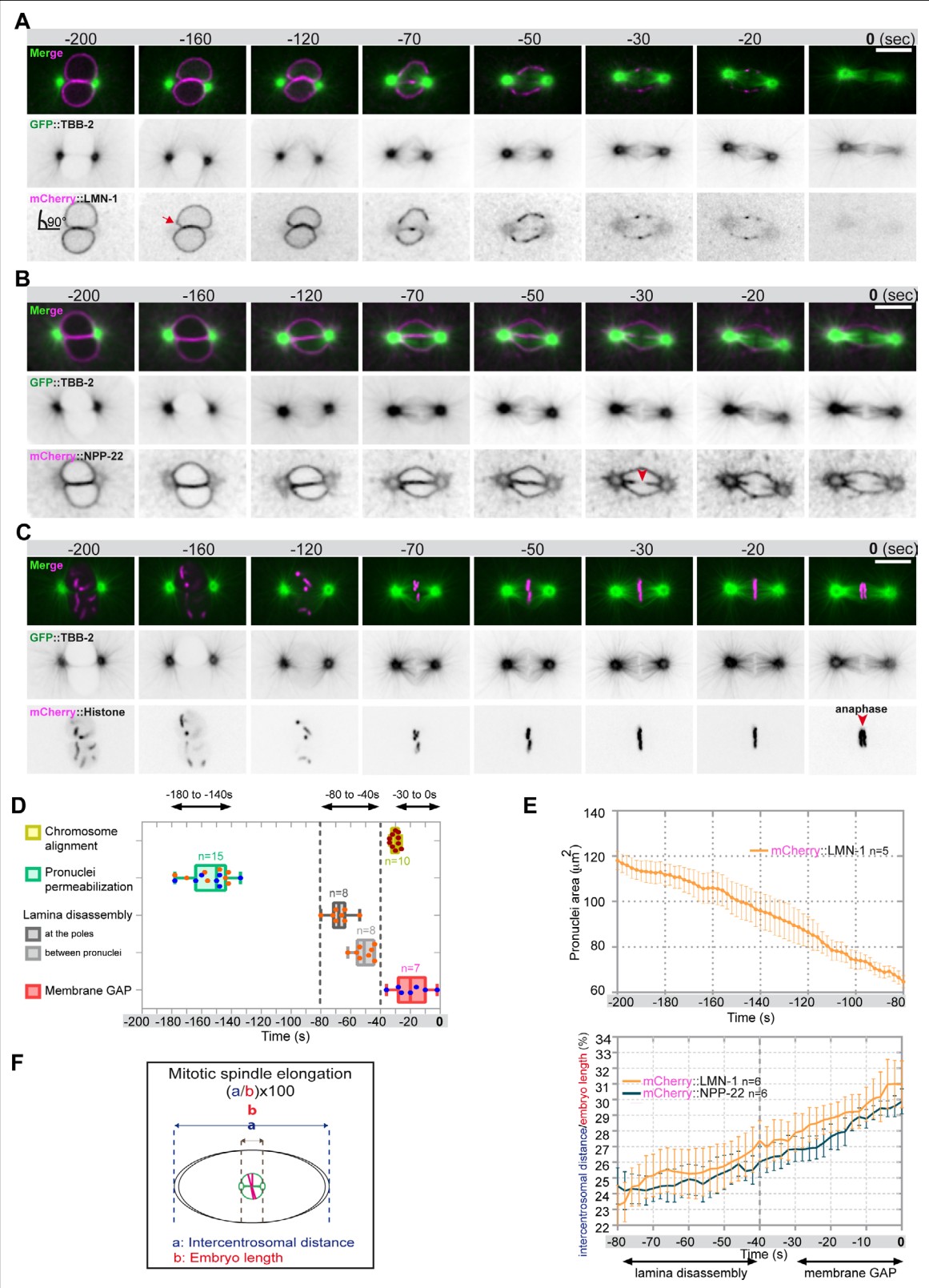

**Figure 1.** Monitoring microtubule and pronuclear envelopes dynamics in the one-cell *Caenorhabditis elegans* embryo. (**A–C**) Spinning disk confocal micrographs of embryos expressing wild-type GFP::TBB-2 (shown alone, and in green in the merged images) and (**A**) mCherry::LMN-1 (shown alone and in magenta in the merged image), or (**B**) mCherry::NPP-22 (magenta, in the merged image), or (**C**) mCherry::Histone (magenta, in the merged image). Timings in seconds are relative to anaphase onset (0 s). All panels are at the same magnification. Scale bar, 10 μm. (**D**) Steps of pronuclear envelopes

*Figure 1 continued on next page*

*Figure 1 continued*

breakdown: pronuclei permeabilization, lamina disassembly (at the poles, between pronuclei), and pronuclear envelopes scission event (membrane gap) relative to anaphase onset were scored in multiple embryos (n). The timing of pronuclear envelopes scission was scored in embryos expressing mCherry::NPP-22 (blue dots, n=7) while the timing of lamina disassembly was scored in embryos expressing mCherry::LMN-1 (orange dots, n=8). (**E**) Graph presenting the surface area occupied by the pronuclei starting 200 s before anaphase onset (0 s) in embryos expressing GFP::TBB-2 and mCherry::LMN-1. ( **F**) Graph presenting the intercentrosomal distance normalized to embryo length starting 80 s before anaphase onset (0 s) in embryos expressing GFP::TBB-2 and mCherry::LMN-1 or mCherry::NPP-22.

The online version of this article includes the following video, source data, and figure supplement(s) for figure 1:

**Source data 1.** Quantification of: (D) The steps of pronuclear envelopes breakdown: pronuclei permeabilization, lamina disassembly (at the poles, between pronuclei), chromosomes alignment, and pronuclear envelopes scission event (membrane gap); (E) The surface area occupied by the pronuclei starting 200 s before anaphase onset (0 s). (F) The intercentrosomal distance normalized to embryo length starting 80 s before anaphase onset (0 s).

**Figure supplement 1.** Timing of pronuclei permeabilization visualized by the appearance of soluble tubulin in the pronuclei.

**Figure supplement 2.** Spatio-temporal lamina depolymerization during mitosis.

**Figure supplement 3.** Timing of pronuclear envelopes scission.

**Figure 1—video 1.** Embryos expressing mCherry::LMN-1 (magenta) and GFP::TBB-2 (green).
https://elifesciences.org/articles/75382/figures#fig1video1

**Figure 1—video 2.** Embryos expressing mCherry::NPP-22 (magenta) and GFP::TBB-2 (green).
https://elifesciences.org/articles/75382/figures#fig1video2

**Figure 1—video 3.** Embryos expressing mCherry::Histone (magenta) and GFP::TBB-2 (green).
https://elifesciences.org/articles/75382/figures#fig1video3

## Paired nuclei phenotype upon reduction of cortical microtubule pulling forces and partial lamina stabilization

The paired nuclei phenotype is a visual readout of the failure to properly remove the nuclear envelope between the pronuclei (*Audhya et al., 2007*; *Bahmanyar et al., 2014*; *Galy et al., 2008*; *Rahman et al., 2015*; *Martino et al., 2017*; *Velez-Aguilera et al., 2020*).

To test whether microtubule pulling forces contribute to pronuclear envelopes breakdown, we examined whether experimental reduction of microtubule-dependent cortical pulling forces could modify the penetrance of the paired nuclei phenotype of embryos expressing the *gfp::lmn-1* 8A allele (*Link et al., 2018*; *Velez-Aguilera et al., 2020*). Because this allele partially stabilizes the lamina, 8% of the embryos expressing GFP::LMN-1 8A present double-paired nuclei at the two-cell stage, and another 9% present a single-paired nuclei cell (*Figure 2A*, *Figure 2—source data 1*). To reduce the cortical pulling forces, we used RNAi to partially deplete GPR-1/2, which are part of an evolutionarily conserved complex anchoring the dynein motor to the embryo cortex (*Colombo et al., 2003*; *Gotta et al., 2003*; *Srinivasan et al., 2003*; *Figure 2B*). Mild depletion of GPR-1/2 in *gfp::lmn-1* 8A embryos greatly enhanced the percentage of embryos presenting a paired nuclei phenotype, with nearly 54% showing double-paired nuclei and 24% a single-paired nuclei (*Figure 2A*, *Figure 2—source data 1*). More severe RNAi-mediated *gpr-1/2* inactivation (Materials and methods) further increased the percentage of *gfp::lmn-1* 8A mutant embryos presenting a double-paired nuclei phenotype to 83% (*Figure 2A*, *Figure 2—source data 1*). Similar treatments of the embryos expressing a wild-type (WT) GFP::LMN-1 allele had only little effect with 5% of embryos presenting a double-paired nuclei phenotype and 4% presenting a single-paired nuclei phenotype (*Figure 2A*, *Figure 2—source data 1*). Thus, reduction of cortical microtubule pulling forces enhances the paired nuclei phenotype of embryos with a partially stabilized lamina network. These observations indicate that microtubule pulling forces facilitate the union of the parental chromosomes in the one-cell embryo.

To test if microtubule pulling forces were directly facilitating lamina disassembly, we monitored GFP::LMN-1 WT and 8A levels throughout mitosis in control versus *gpr-1/2 (RNAi)* embryos, upon partial or severe *gpr-1/2* inactivation. *gpr-1/2* inactivation had no discernable effect on GFP::LMN-1 WT disassembly (*Figure 2C and E*, *Figure 2—source data 1*). Likewise, partial *gpr-1/2* inactivation did not significantly stabilize GFP::LMN-1 8A during mitosis (*Figure 2—figure supplement 1* and *Figure 2—figure supplement 1—source data 1*). However, more severe reduction of cortical microtubule pulling forces, using strong RNAi-mediated *gpr-1/2* inactivation, did stabilize GFP::LMN-1 8A (*Figure 2D and F*, *Figure 2—source data 1*), indicating that microtubule pulling forces contribute to lamina disassembly during mitosis.

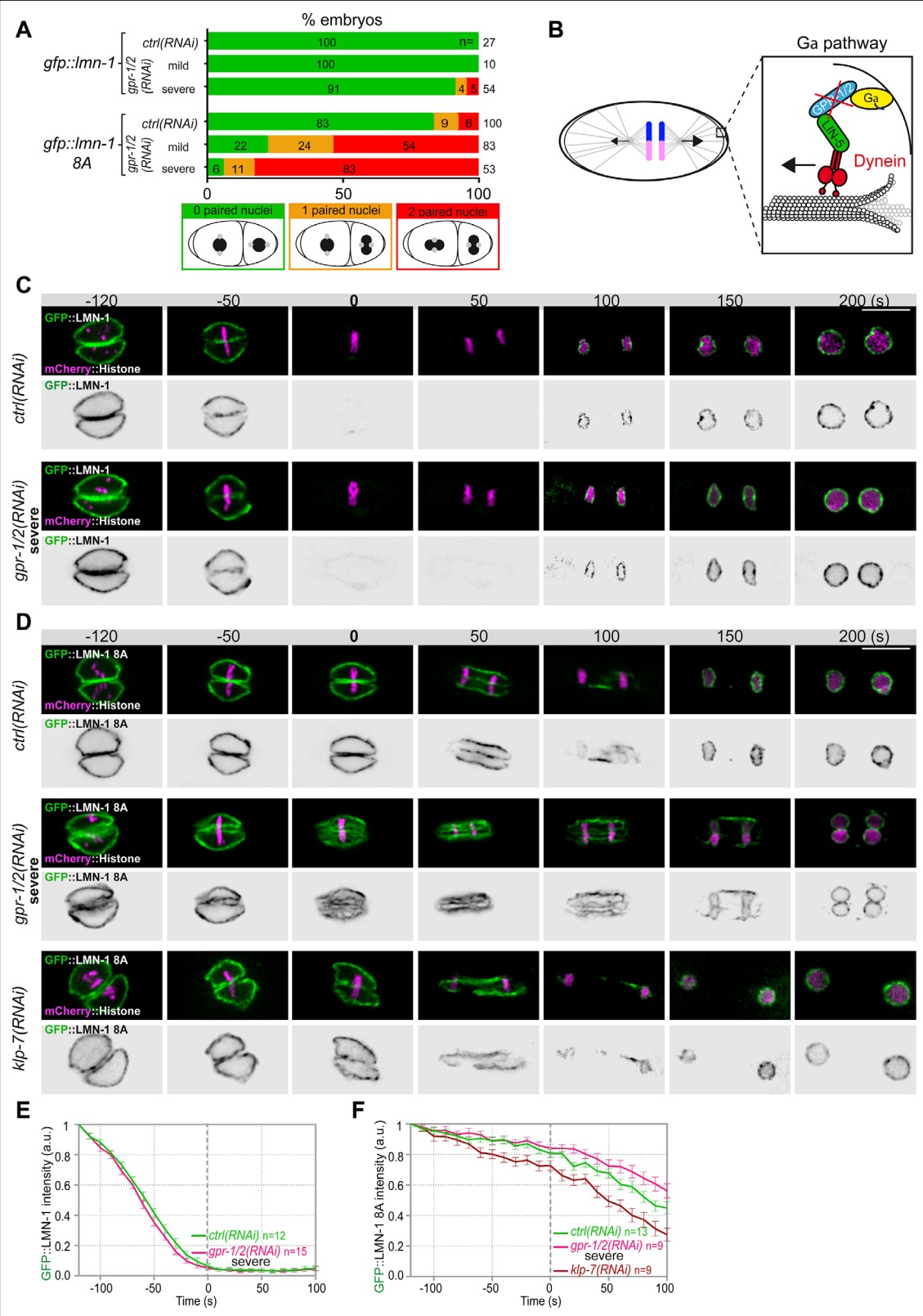

**Figure 2.** Astral microtubule pulling forces contribute to the union of the parental chromosomes during mitosis. (**A**) Percentage of *lmn-1Δ; gfp::lmn-1* and *lmn-1Δ; gfp::lmn-1* 8A embryos presenting 0 (green bars), 1 (orange bars), or 2 (red bars) paired nuclei at the two-cell stage upon exposure to mock RNAi (*ctrl*) or *gpr-1/2(RNAi)* at 20°C. The number of embryos of the indicated phenotype (n) is shown in the graph and was collected from more than three independent experiments. (**B**) Schematics of a one-cell *Caenorhabditis elegans* embryo in anaphase. The astral pulling forces mediated by the Gα

*Figure 2 continued on next page*

*Figure 2 continued*

pathway are schematized in the inset. This pathway, which comprises a complex of Gα (yellow), GPR-1/2 (blue), and LIN-5 (green), anchors dynein (red) to the cell cortex to generate pulling forces when dynein walks toward microtubule minus ends anchored at the spindle poles. Inactivation of GPR-1/2 (red cross) suppresses the astral pulling forces. (**C, D**) Spinning disk confocal micrographs of early *lmn-1Δ* mutant embryos expressing (**C**) wild-type GFP::LMN-1 or (**D**) GFP::LMN-1 8A (shown alone, and in green in the merged images) and mCherry::Histone (magenta, in the merged image) exposed to mock RNAi (*ctrl*) in the upper panels or *gpr-1/2* and *klp-7* RNAi in the lower panels. Times are in seconds relative to anaphase onset (0 s). All panels are at the same magnification Scale bar, 10 μm. (**E, F**) Quantification of (**E**) GFP::LMN-1 or (**F**) GFP::LMN-1 8A fluorescence signal intensity over time from central single focal planes above background at the nuclear envelope in embryos of the indicated genotype during mitosis. Times are in seconds relative to anaphase onset (0 s). The average signal intensity of GFP::LMN-1 and GFP::LMN-1 8A at 120 s before anaphase was defined as 1. The data points on the graphs are the normalized average signal intensity per pixel results ± SEM for n embryos of the indicated genotypes. Data were collected from three independent experiments.

The online version of this article includes the following source data and figure supplement(s) for figure 2:

**Source data 1.** Quantification of: (A) Percentage of *lmn-1Δ; gfp::lmn-1* and *lmn-1Δ; gfp::lmn-1 8A* embryos presenting zero-, one-, or two-paired nuclei at the two-cell stage upon exposure to mock RNAi (*ctrl*) or *gpr-1/2(RNAi)*; (E) GFP::LMN-1 or GFP::LMN-1 8A (F, G) signal intensity during mitosis upon exposure to mock RNAi (*ctrl*), *gpr-1/2(RNAi)*, or *klp-7(RNAi)*.

**Figure supplement 1.** Quantification of GFP::LMN-1 8A signal intensity during mitosis upon reduced astral microtubule pulling forces.

**Figure supplement 1—source data 1.** Quantification of GFP::LMN-1 8A (B) signal intensity during mitosis upon exposure to mock RNAi (*ctrl*) or mild *gpr-1/2(RNAi)*.

To corroborate these observations, we tested whether excessive microtubule pulling forces would facilitate the removal of GFP::LMN-1 8A. To do so, we inactivated the kinesin-13 family member KLP-7, which results in the assembly of an abnormally high number of astral microtubules and thus increases astral cortical pulling forces (*Srayko et al., 2005*; *Gigant et al., 2017*). Loss of *klp-7* caused a premature disassembly of GFP::LMN-1 8A during mitosis (*Figure 2D and F*, *Figure 2—source data 1*), again arguing that astral microtubule pulling forces contribute to the union of the parental chromosomes, at least in part by pulling at the lamina during mitosis. However, our observation that partial inactivation of *gpr-1/2* enhanced the percentage of GFP::LMN-1 8A embryos presenting a paired nuclei phenotype (*Figure 2A*, *Figure 2—source data 1*) without further stabilizing the lamina (*Figure 2—figure supplement 1*, *Figure 2—source data 1*) suggested that cortical microtubule pulling forces might prevent the formation of the paired nuclei phenotype by additional mechanism(s).

## Reducing or increasing cortical microtubule pulling forces affects the timing of membrane scission between the parental pronuclei

Removal of the membranes between the parental chromosomes begins with a membrane scission event right before anaphase onset and after the beginning of mitotic spindle elongation (*Figure 1D and F*). We thus reasoned that by pulling on centrosomes and membranes, and by elongating the mitotic spindle, microtubule pulling forces might mechanically facilitate membrane scission between the parental pronuclei. To investigate this possibility, we used spinning disk confocal microscopy to monitor pronuclear membranes scission upon strong *gpr-1/2* inactivation in embryos expressing the INM protein GFP::LEM-2 and mCherry::Histone, allowing simultaneous visualization of pronuclear envelopes and chromosomes (*Figure 3A*). In control embryos, the membrane scission event between the pronuclei was systematically observed around 30 s before anaphase onset, after chromosomes alignment on the metaphase plate (n=32) (*Figure 3B*, red arrowhead). However, in a vast majority of *gpr-1/2(RNAi)* embryos, the membrane scission between the pronuclei was never detected. Of the 30 embryos analyzed, only 4 presented a membrane scission event (13%) (*Figure 3B and C*, *Figure 3—source data 1*) and these 4 embryos were also the least affected in spindle elongation (*Figure 3D*, *Figure 3—source data 1*), suggesting that spindle elongation contributes to pronuclei membrane scission. Notably, all reforming nuclei were severely misshapen in *gpr-1/2(RNAi)* embryos, but only a fraction of them presented a double-paired nuclei phenotype at the two-cell stage (*Figure 3E*, *Figure 3—source data 1*).

If cortical microtubule pulling forces contribute to the formation of the pronuclei membrane scission event by elongating the mitotic spindle, excessive pulling forces might induce a premature membrane scission relative to anaphase onset. To test this hypothesis, we used two complementary approaches to increase astral microtubule pulling forces. We inactivated *klp-7* as before, or *efa-6*

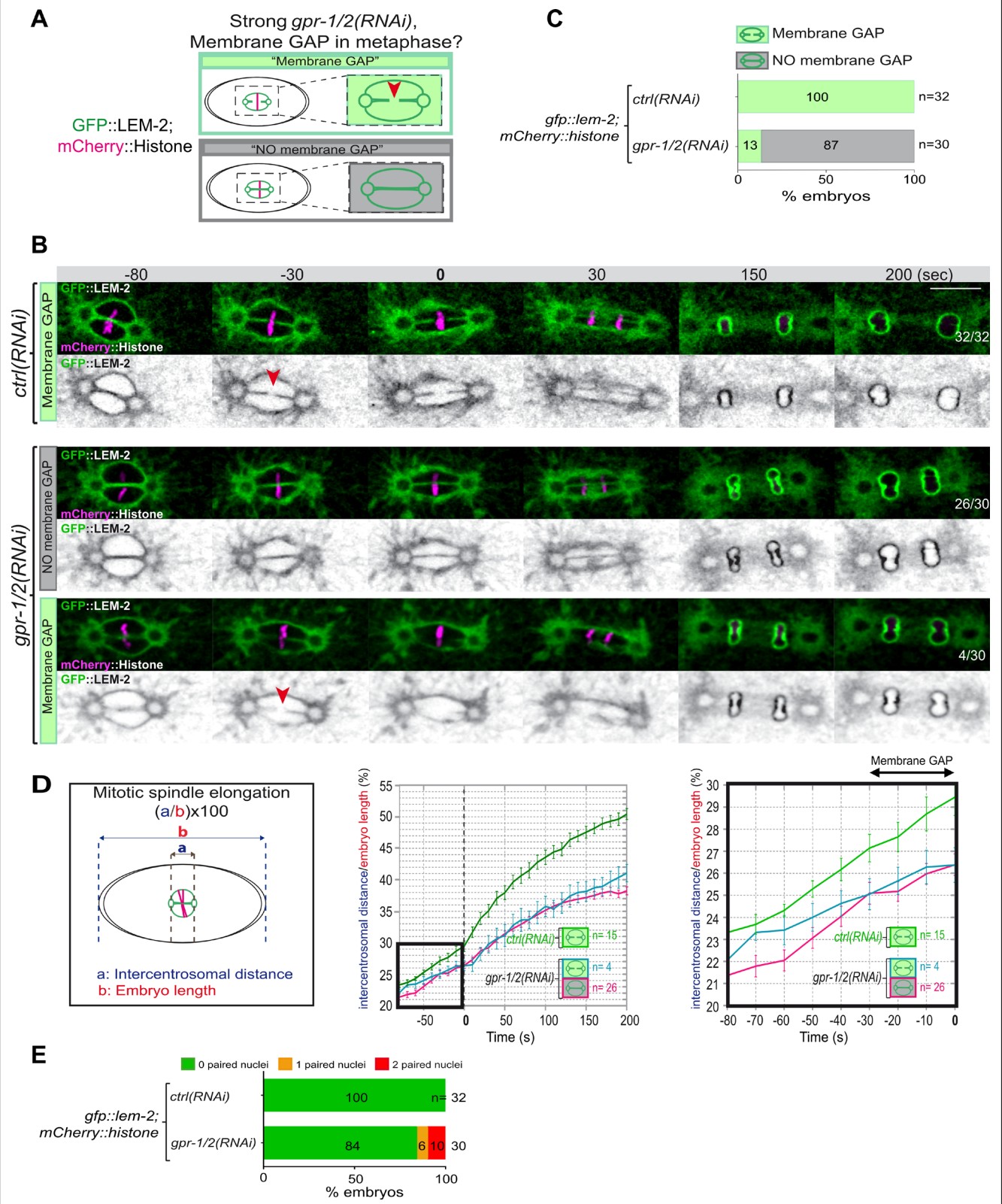

**Figure 3.** Reduction of astral pulling forces prevents pronuclear membranes scission. (**A**) Schematics of the approach used to test the effect of a reduction of astral microtubule pulling forces on pronuclear envelopes scission (membrane gap) during mitosis. (**B**) Spinning disk confocal micrographs of one-cell stage embryos expressing the inner nuclear membrane protein GFP::LEM-2 (shown alone, and in green in the merged images) and mCherry::Histone (magenta, in the merged image) exposed to mock RNAi (*ctrl*) in the upper panels or *gpr-1/2(RNAi)* in the lower panels. Times

*Figure 3 continued on next page*

*Figure 3 continued*

are in seconds relative to anaphase onset (0 s). The fraction of embryos that showed the presented phenotype is indicated at the bottom right of each image. All panels are at the same magnification Scale bar, 10 μm. The orange arrowheads point to the pronuclear envelopes scission event. (**C**) Percentage of embryos presenting a pronuclear envelopes scission event upon exposure to mock RNAi (*ctrl*) or *gpr-1/2(RNAi)*. (**D**) Graphs showing the intercentrosomal distance normalized to embryo length in percentage during mitosis upon exposure to mock RNAi (*ctrl*) or *gpr-1/2(RNAi)*. Times are in seconds relative to anaphase onset (0 s). (**E**) Percentage of embryos presenting zero- (green bars), one- (orange bars), or two- (red bars) paired nuclei at the two-cell stage upon exposure to mock RNAi (*ctrl*) or *gpr-1/2(RNAi)*. The number of embryos analyzed (n) is indicated on the graph and was collected from three independent experiments.

The online version of this article includes the following source data for figure 3:

**Source data 1.** Quantification of: (C) Percentage of embryos presenting a pronuclear envelopes scission event; (D) The intercentrosomal distance normalized to embryo length starting 80 s before anaphase onset (0 s); (E) Percentage of *gfp-lem-2* embryos presenting zero-, one-, or two-paired nuclei at the two-cell stage upon exposure to mock RNAi (*ctrl*) or *gpr-1/2(RNAi)*.

(exchange factor for Arf), which encodes a cortically localized protein that limits the growth of microtubules near the cell cortex of early embryonic cells. Loss of EFA-6 causes excess centrosome separation and displacement toward the cell cortex early in mitosis and subsequently increased rates of spindle elongation (*O'Rourke et al., 2010*). We inactivated *klp-7* or *efa-6* in embryos expressing GFP::LEM-2 and mCherry::Histone to monitor pronuclear membranes scission relative to anaphase onset and mitotic spindle length (*Figure 4A*). While the control embryos underwent membrane scission between the juxtaposed pronuclei 0–30 s before anaphase onset, this event occurred systematically earlier in *efa-6* and *klp-7(RNAi)* embryos. In 50% of these embryos, pronuclear membranes scission occurred between 40 and 120 s before anaphase onset (*Figure 4B and C*, *Figure 4—source data 1*). Membrane scission occurred earlier in these embryos as a consequence of premature mitotic spindle elongation (*Figure 4D*, *Figure 4—source data 1*). By measuring mitotic spindle length at the time of membrane gap formation, we noticed that membranes scission systematically occurred at a similar mitotic spindle length (*Figure 4E*, *Figure 4—source data 1*). Taken together, these observations indicate that microtubule pulling forces, by promoting mitotic spindle elongation, contribute to pronuclear membranes scission, possibly by tearing apart the pronuclear membranes.

## Lamina depolymerization and chromosome alignment are prerequisites for membrane gap formation even in the presence of excessive pulling forces

Embryos with a stabilized lamina, expressing the non-phosphorylable LMN-1 8A, are systematically defective in pronuclear envelopes scission (*Velez-Aguilera et al., 2020*). We reasoned that the lamina when stabilized, in addition to constituting a physical barrier between chromosomes, could oppose the pulling forces exerted by astral microtubules and could thus prevent elongation of the mitotic spindle during anaphase. To test this model, we measured mitotic spindle elongation in WT versus *lmn-1* 8A mutant embryos by measuring the intercentrosomal distance during mitosis. In contrast to WT, the spindle did not elongate to the same extent in *lmn-1* 8A mutant embryos (*Figure 5A*, *Figure 5—source data 1*), indicating that stabilization of the lamina interferes with mitotic spindle elongation and thus that lamina depolymerization facilitates spindle elongation.

We then asked whether excessive microtubule pulling forces might rescue pronuclear envelopes scission in embryos defective in lamina depolymerization. To test this hypothesis, we inactivated *klp-7* in *lmn-1* 8A embryos expressing GFP::LEM-2 and mCherry::Histone and monitored the nuclear envelope dynamics and mitotic spindle elongation by spinning disk confocal microscopy. While these treatments rescued mitotic spindle elongation defects of *lmn-1* 8A embryos (*Figure 5B*, *Figure 5—source data 1*), they failed to restore the formation of pronuclear membranes scission and single nucleus embryos at the two-cell stage (*Figure 5C and D*).

These observations indicate that even premature and excessive spindle elongation is not sufficient to induce membrane scission when the lamina is stabilized during mitosis.

Collectively, these observations indicate that lamina depolymerization is a prerequisite for pronuclear envelopes scission.

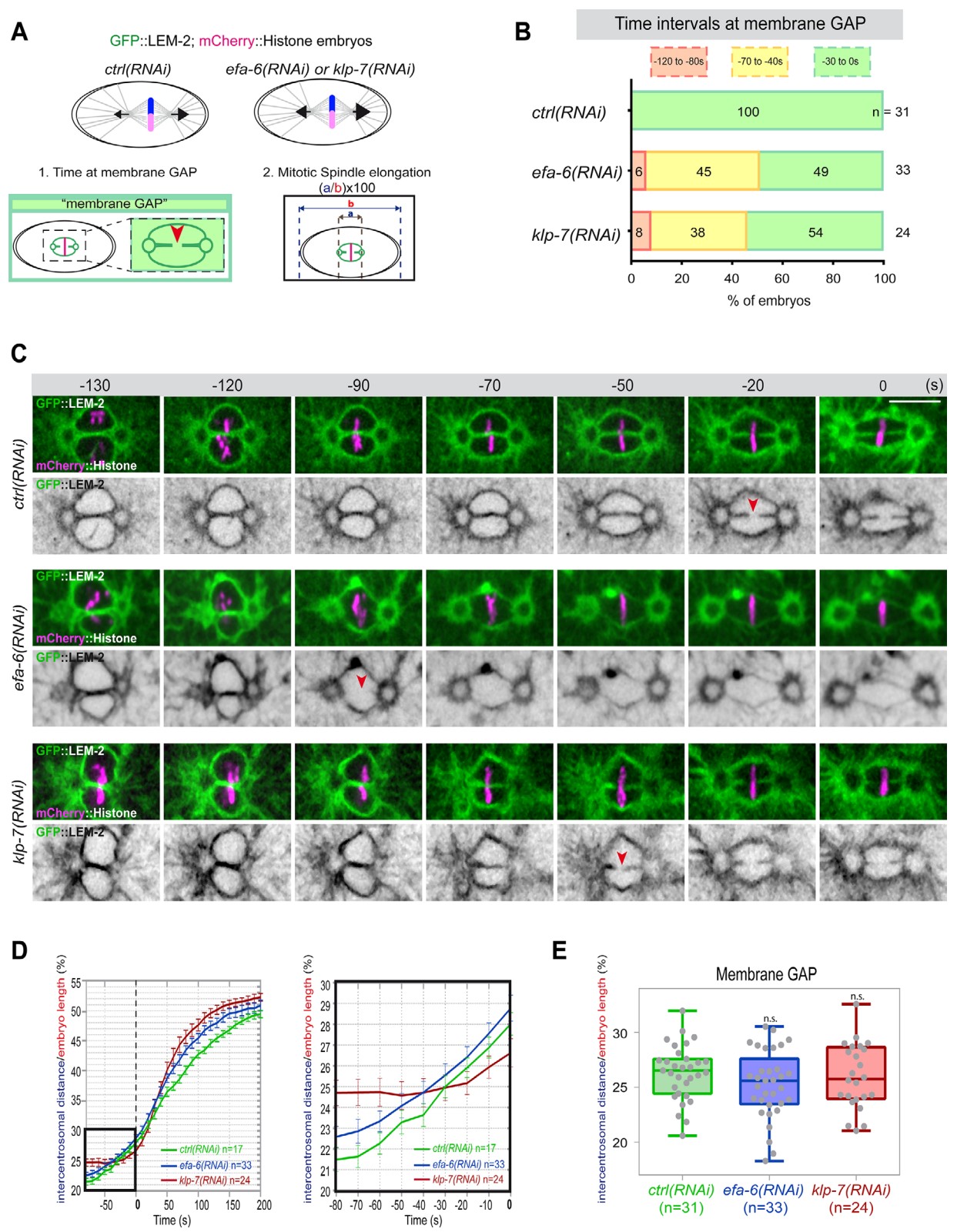

**Figure 4.** Premature pronuclear membranes scission event upon excessive astral microtubule pulling forces. (**A** ) Schematics of the approach to test the effect of excessive astral microtubule pulling forces on pronuclear membranes scission (membrane gap) (1) and mitotic spindle elongation (2). (**B**) Percentage of embryos presenting the pronuclear membranes scission event at different time intervals relative to anaphase onset (0 s). The number of embryos (n) analyzed is indicated on the graph and was collected from three independent experiments. (**C–**) Spinning disk confocal micrographs

*Figure 4 continued on next page*

*Figure 4 continued*

of early embryos expressing mCherry::Histone, GFP::LEM-2 exposed to mock RNAi (*ctrl*), *efa-6(RNAi)*, or *klp-7(RNAi)*. Times are in seconds relative to anaphase onset (0 s). The red arrowheads indicate pronuclear membranes scission. All panels are at the same magnification Scale bar, 10 μm. (**D**) Intercentrosomal distance normalized to embryo length in percentage during mitosis upon exposure to mock RNAi (*ctrl*), *efa-6*, or *klp-7(RNAi)*. Times are in seconds relative to anaphase onset (0 s). The graph on the right is a zoom of the first graph focused on the 80 s before anaphase onset (0 s). (**E**) Box and Whisker plots presenting the intercentrosomal distance normalized to embryo length in percentage at the time of pronuclear membranes gap formation in embryos of the indicated genotypes. n=number of embryos analyzed.

The online version of this article includes the following source data and figure supplement(s) for figure 4:

**Source data 1.** Quantification of: (B) Percentage of embryos presenting a pronuclear membranes scission event at different time intervals relative to anaphase onset (0s); (D) The intercentrosomal distance normalized to embryo length starting 80s before anaphase onset (0s); (E) The intercentrosomal distance normalized to embryo length in percentage at the time of pronuclear membranes gap formation in embryos upon exposure to mock RNAi (*ctrl*), *efa-6(RNAi)*, or *klp-7(RNAi)*.

**Figure supplement 1.** Chromosome configuration at the time of membrane gap formation upon an excessive astral microtubule pulling forces.

Previous work has shown that a defect in chromosome alignment on the metaphase plate prevents pronuclear membranes scission (*Rahman et al., 2015*). In the *efa-6* or *klp-7(RNAi)* embryos with enhanced astral pulling forces, pronuclear membranes scission often occurred before chromosomes were fully aligned on the metaphase plate (*Figure 4—figure supplement 1*) suggesting that membrane scission between the pronuclei may not require full and complete chromosome alignment on the metaphase plate.

To further investigate this possibility, we examined whether excessive pulling forces can trigger pronuclear envelopes scission between the pronuclei in the total absence of chromosome alignment. To this end, we inactivated *klp-7* and the essential kinetochore protein HCP-3[CENPA] to prevent chromosome congression on the metaphase plate (*Oegema et al., 2001*). In these embryos, no membrane scission was detected despite the extensive mitotic spindle elongation (*Figure 5C*). These observations indicate that chromosome localization in the vicinity of the pronuclear membranes is necessary for pronuclear membranes scission and might dictate the site of membranes scission.

## The essential role of PLK-1 in pronuclear envelopes scission is to promote lamina depolymerization

PLK-1 is critically required for pronuclear envelopes scission (*Rahman et al., 2015*; *Martino et al., 2017*). Accordingly, membranes scission is prevented in *plk-1*ts embryos. We thus investigated the exact role of PLK-1 in pronuclear envelopes scission. We previously showed that LMN-1 is a key PLK-1 target in this process as the sole expression of the non-phosphorylable *lmn-1* 8A allele is sufficient to prevent pronuclear envelopes scission (*Velez-Aguilera et al., 2020*). Whether PLK-1 regulates pronuclear envelopes scission by targeting other substrates is currently unclear. For instance, PLK-1 could regulate membrane scission by activating a factor essential for pronuclear envelopes scission. Recent Focused Ion Beam-Scanning Electron Microscopy (FIB-SEM) analysis has revealed that the four membranes of the pronuclei fuse and become two via a novel membrane structure, the three-way sheet junctions (*Rahman et al., 2020*). These junctions are absent in *plk-1*ts embryos (*Rahman et al., 2020*) raising the possibility that PLK-1 could directly regulate their formation.

To discriminate between these hypotheses, we asked whether partial *lmn-1* inactivation by RNAi is sufficient to restore pronuclear envelopes scission in *plk-1*ts embryos (*Figure 6A*). If *lmn-1* inactivation in *plk-1*ts does not restore pronuclear envelopes scission, this would argue that PLK-1 has, beyond lamina depolymerization, other roles to promote membrane gap formation. However, if *lmn-1* inactivation is sufficient to restore membrane gap formation in *plk-1*ts, this would suggest that LMN-1 is possibly the only key PLK-1 target in this process, unless partial *lmn-1* inactivation indirectly affects an inhibitor of pronuclear envelopes scission.

To monitor pronuclear envelopes scission in *plk-1*ts mutant embryos, we constructed a *plk-1*ts strain co-expressing GFP::LEM-2 and mCherry::EMR-1 (EMERIN), which both localize to the inner nuclear envelope and directly interact with the lamina (*Figure 6A*). We then used

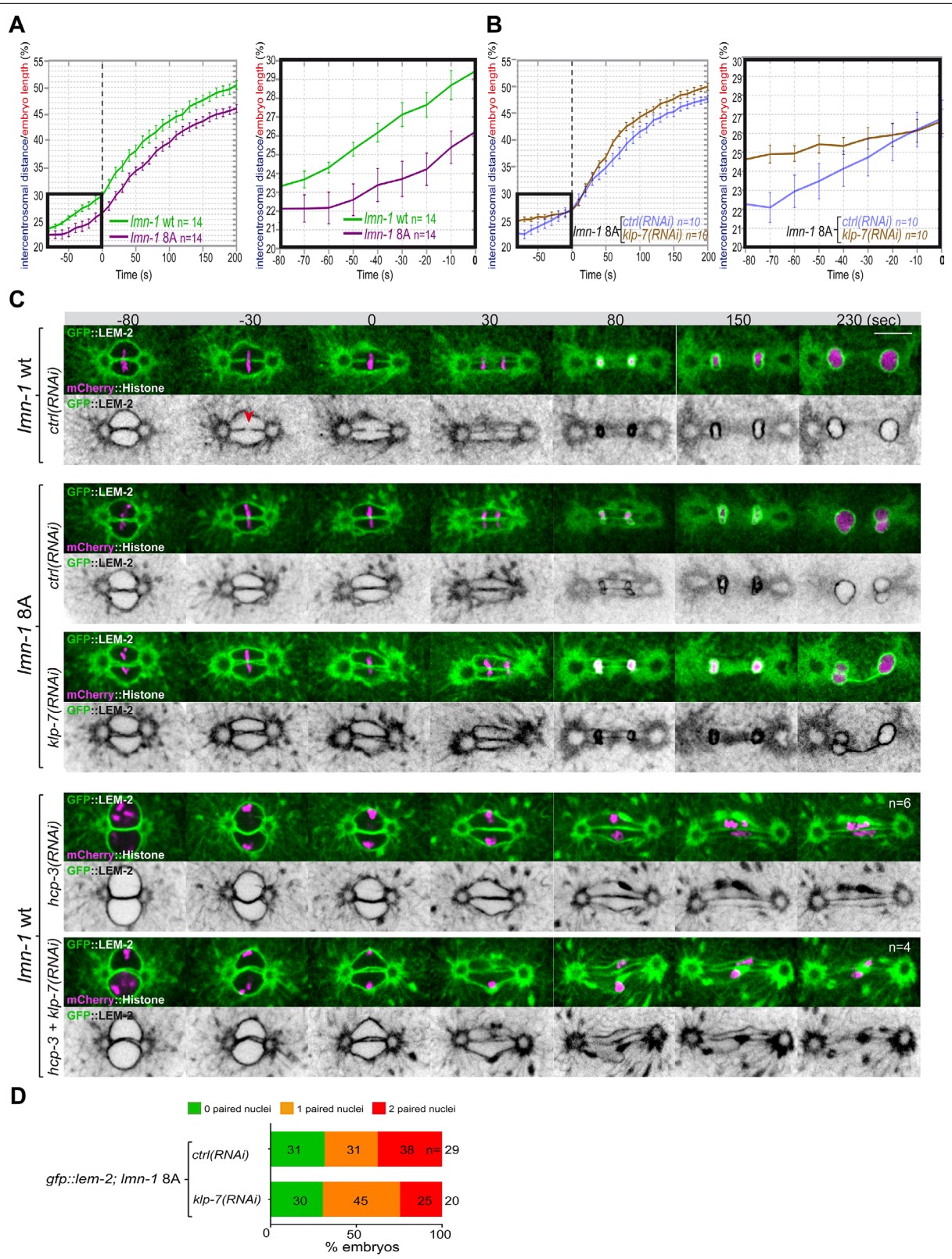

**Figure 5.** Chromosome alignment and lamina depolymerization are prerequisites for pronuclear membranes scission. (**A**) Graphs showing the intercentrosomal distance normalized to embryo length in percentage in wild-type and *lmn-1* 8A embryos during mitosis. Times are in seconds relative to anaphase onset (0 s). The graph on the right is a zoom of the first graph focused on the 80 s before anaphase onset (0 s). (**B**) Graphs showing the intercentrosomal distance normalized to embryo length in percentage in *lmn-1* 8A embryos exposed to mock RNAi (*ctrl*) or *klp-7(RNAi)* during mitosis.

*Figure 5 continued on next page*

*Figure 5 continued*

Times are in seconds relative to anaphase onset (0 s). The graph on the right is a zoom of the first graph focused on the 80 s before anaphase onset (0 s). (**C**) Spinning disk confocal micrographs of one-cell stage embryos of the indicated genotype expressing the inner nuclear membrane protein GFP::LEM-2 (shown alone, and in green in the merged images) and mCherry::Histone (magenta, in the merged image). Times are in seconds relative to anaphase onset (0 s). The fraction of embryos that showed the presented phenotype is indicated at the bottom right of each image. All panels are at the same magnification Scale bar, 10 µm. The orange arrowheads point to the pronuclear envelopes scission event. (**D**) Percentage of embryos presenting zero- (green bars), one- (orange bars), or two- (red bars) paired nuclei at the two-cell stage upon exposure to mock RNAi (*ctrl*) or *klp-7(RNAi)*. The number of embryos analyzed (n) is indicated on the graph and was collected from three independent experiments.

The online version of this article includes the following source data for figure 5:

**Source data 1.** Quantification of: (A) The intercentrosomal distance normalized to embryo length starting 80 s before anaphase onset (0 s) in *lmn-1* wt and *lmn-1* 8A embryos; (B) The intercentrosomal distance normalized to embryo length starting 80s before anaphase onset (0s) in *lmn-1* 8A embryos upon exposure to mock (control, ctrl) or *klp-7(RNAi)*; (E) Percentage of *gfp-lem-2; lmn-1* 8A embryos presenting zero-, one-, or two-paired nuclei at the two-cell stage upon exposure to mock RNAi (*ctrl*) or *gpr-1/2(RNAi)*.

spinning disk confocal microscopy to monitor pronuclear membranes configuration in one-cell embryos. As reported previously, pronuclear envelopes scission between the pronuclei was totally prevented in *plk-1*ts embryos at restrictive temperature (*Rahman et al., 2020*; *Rahman et al., 2015*). However, partial RNAi-mediated *lmn-1* inactivation was sufficient to restore pronuclear envelopes scission (*Figure 6B*). *lmn-1* inactivation also suppressed the paired nuclei phenotype of *plk-1*ts embryos in these conditions (*Figure 6C*, *Figure 6—source data 1*), as reported previously (*Rahman et al., 2015*; *Martino et al., 2017*). *lmn-1* inactivation not only restored membrane scission and the shape of the pronuclei, but also the normal localization of LEM-2 and EMR-1.

In *plk-1*ts embryos, pronuclei appeared rounded, likely due to the persisting lamina that provides rigidity to the nuclear membrane. GFP::LEM-2 and mCherry::EMR-1 readily accumulated on the nuclear envelope. *lmn-1* inactivation restored the triangular shape of the mitotic spindle and the correct localization of LEM-2 and EMR-1 to the endoplasmic reticulum surrounding the centrosomes (*Figure 6A, B and C*). We speculate that by fluidifying the pronuclear membranes, lamina depolymerization might facilitate the release of INM proteins back into the peripheral ER surrounding the centrosomes.

Collectively, these observations suggest that LMN-1 might be the only PLK-1 direct target involved in regulating pronuclear membranes scission.

In conclusion, our study indicates that cortical microtubule pulling forces are required for pronuclear membranes scission but are not sufficient when chromosomes are not aligned on the metaphase plate, or when the lamina persists between the parental chromosomes. Coordination between chromosome alignment, lamina depolymerization, and mitotic spindle elongation is thus required for pronuclear envelopes scission in the early *C. elegans* embryo (*Figure 6D*). In addition, our results indicate that a lack of membrane scission does not systematically result in a paired nuclei phenotype. While *gpr-1/2* inactivation prevents membrane scission, most reforming nuclei were misshapen but not double paired at the two-cell stage. One possible explanation for this observation is that the lamina disassembles in *gpr-1/2(RNAi)* embryos, which results in membrane flexibility, and allows integral membrane proteins to redistribute into the peripheral ER. Then, during nuclear envelopes reformation and expansion, the membrane separating the parental chromosomes is weak and eventually dismantles, which may explain why single pronuclei manage to form at the two-cell stage in *gpr-1/2(RNAi)* embryos. It is noteworthy that in all reported conditions where a correlation between a lack of membrane scission and the paired nuclei phenotype was observed, depolymerization of the lamina was systematically delayed or inhibited (*Audhya et al., 2007*; *Galy et al., 2008*; *Golden et al., 2009*; *Bahmanyar et al., 2014*; *Rahman et al., 2015*; *Martino et al., 2017*; *Velez-Aguilera et al., 2020*). Persistence of the lamina rigidifies the nuclear membrane, prevents the redistribution of integral membrane proteins into the ER, and counteracts the mitotic spindle elongation, which typically results in a penetrant paired nuclei phenotype.

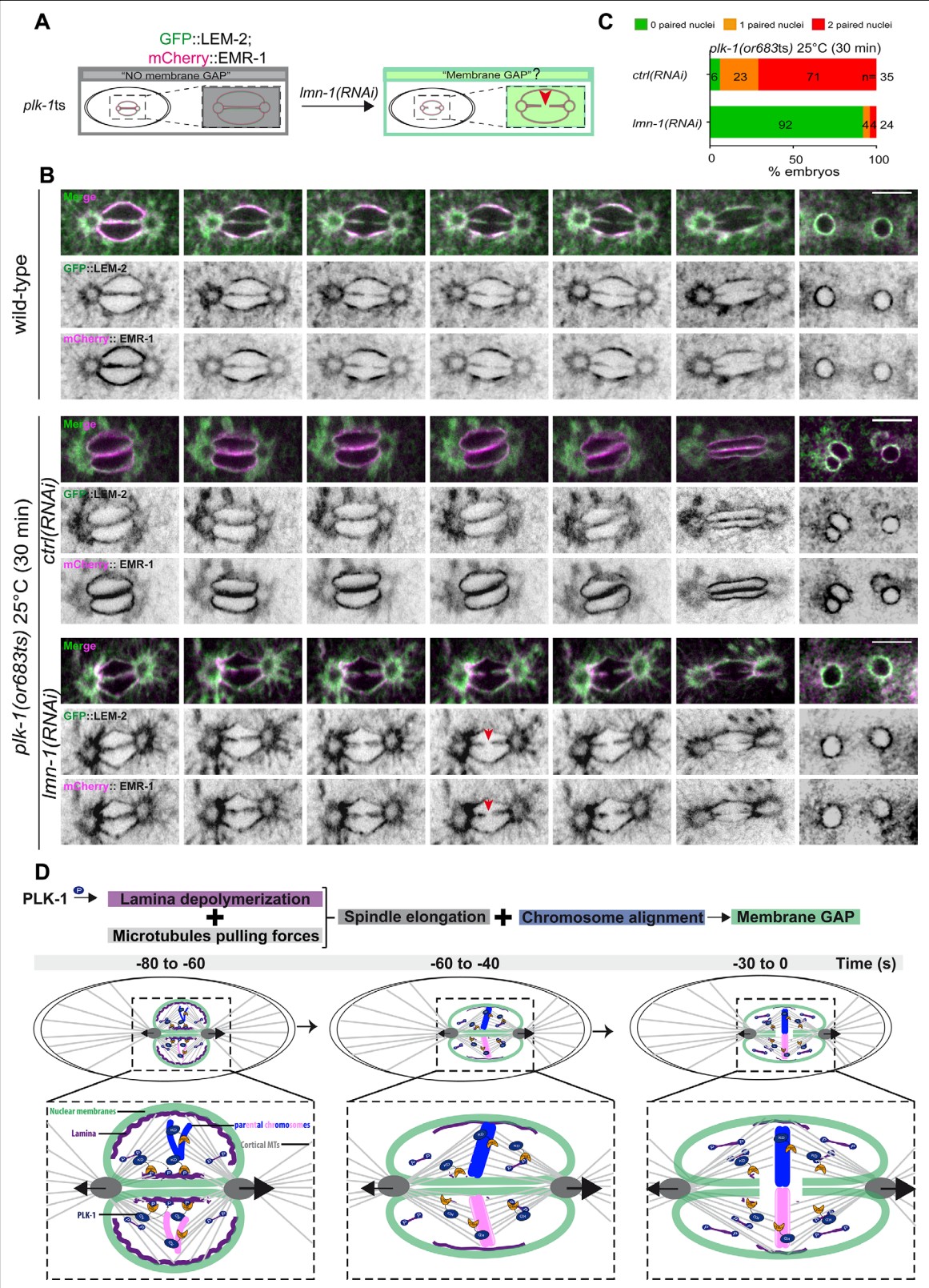

**Figure 6.** PLK-1 triggers pronuclear membranes scission mainly via lamina depolymerization. (**A**) The schematics present the approach used to test whether partial *lmn-1* inactivation in *plk-1*ts embryos expressing GFP::LEM-2 and mCherry::EMR-1 is sufficient to restore membrane gap formation. (**B**) Spinning disk confocal micrographs of wild-type or *plk-1*ts early embryos expressing mCherry::EMR-1 and GFP::LEM-2 exposed mock RNAi (*ctrl*) or *lmn-1(RNAi)*. All panels are at the same magnification. Scale bar, 10 µm. (**C**) Percentage of embryos presenting zero- (green bars), one- (orange bars), or

*Figure 6 continued on next page*

*Figure 6 continued*

two- (red bars) paired nuclei at the two-cell stage upon exposure to mock RNAi (*ctrl*) or *lmn-1(RNAi)*. The number of embryos analyzed (n) is indicated on the graph and was collected from three independent experiments. (**D**) Working model: temporal coordination between lamina depolymerization, chromosome alignment, and mitotic spindle elongation is required for pronuclear envelopes scission and parental genomes unification in the early *Caenorhabditis elegans* embryo.

The online version of this article includes the following source data for figure 6:

**Source data 1.** Quantification of: (C) Percentage of *plk-1(or683ts)* embryos, expressing GFP::LEM-2, and mCherry::EMR-1 presenting zero-, one-, or two-paired nuclei at the two-cell stage upon exposure to mock RNAi (*ctrl*) or *lmn-1(RNAi)*.

Finally, our observations that a defective lamina depolymerization interferes with mitotic spindle elongation highlights the role of nuclear envelope remodeling in influencing mitotic spindle length.

# Materials and methods

## Key resources table

| Reagent type (species) or resource | Designation | Source or reference | Identifiers | Additional information |
|---|---|---|---|---|
| Strain, strain background (*Caenorhabditis elegans*) | C. elegans N2 Bristol | *Caenorhabditis* Genetics Center (CGC) | http://www.cgc.cbs.umn.edu/strain.php?id=10570 | |
| Strain, strain background (*C. elegans*) | mCherry::NPP-22 *npp-22(syb1474)V* | SunyBiotech | PHX1774 | |
| Strain, strain background (*C. elegans*) | mCherry::LMN-1; GFP::TBB-2: [*mCherry::lmn-1*] MosSCI: *jfSi68[lmn-1(4 kb 5'UTR)::mCherry::lmn-1gDNA exon four recoded::3'UTR lmn-1 cb-unc-119(+)]II; ojIs1[Ppie-1_gfp::tbb-2]; unc-119(ed3)III* | Pintard lab This study | WLP996 | |
| Strain, strain background (*C. elegans*) | mCherry::NPP-22; GFP::TBB-2:*ojIs1[Ppie-1_gfp::tbb-2, cb-unc-119(+)]; unc-119(ed3)III; mCherry::npp-22(syb1474)V* | Pintard lab This study | WLP993 | |
| Strain, strain background (*Caenorhabditis elegans*) | mCherry::HIS-58; GFP::TBB-2:*ojIs1[Ppie-1_gfp::tbb-2]; unc-119(ed3)III; ltIs37[pAA64; Ppie1_mCherry::his-58; unc-119 (+)]IV* | CGC | JCC483 | |
| Strain, strain background (*C. elegans*) | GFP::LMN-1; mCherry::Histone:*lmn-1(tm1502)I; jfSi68[Plmn-1::gfp cb-unc-119(+)]II; mCherry::his-58 IV* | *Link et al., 2018* | UV142 | |
| Strain, strain background (*C. elegans*) | GFP::LMN-1 8A; mCherry::Histone:*lmn-1(tm1502)I;jfSi89[Plmn-1S(21,22,24,32,397,398,403, 405)A::gfp cb-unc-119(+)]II; mCherry::his-58 IV* | *Link et al., 2018* | UV144 | |
| Strain, strain background (*C. elegans*) | GFP::LEM-2; mCherry::HIS-58: *ltIs37 [(pAA64) pie-1p::mCherry::his-58 + unc-119(+)]IV qals3507 [pie-1::GFP::LEM-2 + unc-119(+)]* | CGC | OD83 | |
| Strain, strain background (*C. elegans*) | lmn-1 8A; GFP::LEM-2; mCherry::HIS-58:*lmn-1^S8A S(21,22,24,32,397,398,403, 405)A ltIs37 [(pAA64) pie-1p::mCherry::his-58 + unc-119(+)]IV qals3507 [pie-1::GFP::LEM-2 + unc-119(+)]* | *Velez-Aguilera et al., 2020* | WLP833 | |
| Strain, strain background (*C. elegans*) | GFP::LEM-2; mCherry::EMR-1:*bqSi210 [lem-2p::lem-2::GFP + unc-119(+)] II; bqSi226 [emr-1p::emr-1::mCherry + unc-119(+)]IV* | *Morales-Martínez et al., 2015* | BN228 | |
| Strain, strain background (*C. elegans*) | plk-1(or683ts); GFP::LEM-2; mCherry::EMR-1:*bqSi210 [lem-2p::lem-2::GFP+unc-119(+)] II; plk-1(or683ts)III; bqSi226 [emr-1p::emr-1::mCherry + unc-119(+)] IV* | Pintard lab This study | WLP1041 | |
| Strain, strain background (*Escherichia coli*) | OP50 | CGC | http://www.cgc.cbs.umn.edu/strain.php?id=11078 | |
| Strain, strain background (*E. coli*) | HT115(DE3) | CGC | http://www.cgc.cbs.umn.edu/strain.php?id=11078 | |

*Continued on next page*

*Continued*

| Reagent type (species) or resource | Designation | Source or reference | Identifiers | Additional information |
|---|---|---|---|---|
| Chemical compound, drug | IPTG | Euromedex | Cat#EU0008-B | |
| Chemical compound, drug | Pfu | Promega | Cat#M7741 | |
| Chemical compound, drug | DpnI | Biolabs | Cat#R0176S | |
| Commercial assay or kit | BP Clonase II Enzyme Mix (Gateway cloning) | Invitrogen | Cat#11789-020 | |
| Commercial assay or kit | LR Clonase II Enzyme Mix (Gateway cloning) | Invitrogen | Cat#11791-020 | |
| Recombinant DNA reagent | L4440 (RNAi Feeding vector) | *Kamath et al., 2001* | N/A | |
| Recombinant DNA reagent | *gpr-1/2* cloned into L4440 | *Kamath et al., 2003* | Arhinger Library | |
| Recombinant DNA reagent | *klp-7* cloned into L4440 | *Kamath et al., 2003* | Arhinger Library | |
| Recombinant DNA reagent | *efa-6* cloned into L4440 | *Kamath et al., 2003* | Arhinger Library | |
| Recombinant DNA reagent | *hcp-3* cloned into L4440 | *Kamath et al., 2003* | Arhinger Library | |
| Recombinant DNA reagent | *lmn-1* cloned into L4440 | *Kamath et al., 2003* | Arhinger Library | |
| Recombinant DNA reagent | pDESTttTi5605[R4-R3] for MOS insertion on Chromosome II | *Frøkjaer-Jensen et al., 2008* | pCFJ150 Addgene plasmid # 19329 | |
| Recombinant DNA reagent | MOS transposase Pglh-2::MosTase::glh-2utr | *Frøkjaer-Jensen et al., 2008* | pJL43.1 Addgene plasmid # 19332 | |
| Recombinant DNA reagent | *Prab-3::mCherry* | *Frøkjaer-Jensen et al., 2008* | pGH8 | |
| Recombinant DNA reagent | *Pmyo-2::mCherry::unc-54* | *Frøkjaer-Jensen et al., 2008* | pCFJ90 | |
| Recombinant DNA reagent | Plmn-1_gfp::lmn-1_lmn-1 3'UTR in pCFJ150 | *Link et al., 2018* | N/A | |
| Recombinant DNA reagent | Plmn-1_gfp::lmn-1(S8A) lmn-1 3'UTR in pCFJ150 | *Link et al., 2018* | N/A | |
| Recombinant DNA reagent | mCherry::lmn-1 in pCFJ150 | This study | pLP2437 | |
| Sequence-based reagent | Forward to amplify 5' of mCherry | This study | OLP2570 PCR primers | CTCTTCAGAAAGCAGCGAGA AAAATGGGA GGTAGGGCCGGCTCTG |
| Sequence-based reagent | Reverse to amplify 5' of mCherry | This study | OLP2571 PCR primers | CAGAGCCGGCCCTACCTCCC ATTTTTCT CGCTGCTTTCTGAAGAG |
| Sequence-based reagent | Forward to amplify 3' of mCherry with linker before LMN-1 | This study | OLP2572 PCR primers | GGTGGCATGGATGAATTGTA TAAGGCAAGT TTGTACAAAAAAGCAGGCTCC |
| Sequence-based reagent | oJD580 Amp_For to amplify fragment of PCFJ150 | This study | OLP870 PCR primers | ATCGTGGTGTCACGCTCGTC GTTTGGTATGG |
| Sequence-based reagent | oJD581 Amp_Rev to amplify fragment of PCFJ151 | This study | OLP871 PCR primers | ATACCAAACGACGAGCGTGA CACCACGATGC |
| Sequence-based reagent | Gibson Forward oligo for MosII LMN-1 construction. | This study | OLP2267 | CCTTGTCCGAATCCACCACC CATTCCTCCTG |

*Continued on next page*

*Continued*

| Reagent type (species) or resource | Designation | Source or reference | Identifiers | Additional information |
|---|---|---|---|---|
| Sequence-based reagent | Gibson Reverse oligo for MosII LMN-1 construction. | This study | OLP2266 | GGAGGAATGGGTGGTGGATT CGGACAAGGAC |
| Software, algorithm | Adobe Illustrator CS6 | Adobe | https://www.adobe.com/products/illustrator.html | |
| Software, algorithm | Adobe Photoshop CS4 | Adobe | https://www.adobe.com/products/photoshop.html | |
| Software, algorithm | ImageJ | NIH *Schneider et al., 2012* | https://imagej.nih.gov/ij/ | |
| Software, algorithm | ZEN | Zeiss | https://www.zeiss.com/microscopy/int/products/microscope-software/zen.html | |
| Software, algorithm | PRISM | Graphpad | https://www.graphpad.com/ | |
| Software, algorithm | Metamorph | Molecular Devices | https://www.metamorph.com/ | |
| Software, algorithm | Imaris | Bitplane | Microscopy Image Analysis Software - Imaris - Oxford Instruments (oxinst.com) | |

## Contact for reagent and resource sharing

Further information and requests for reagents may be directed to and will be fulfilled by the lead contact author L Pintard: lionel.pintard@ijm.fr.

## Experimental model and subject details

*C. elegans* and bacterial strains used in this study are listed in the Key resources table.

## Method details

### Molecular biology

The plasmids and oligonucleotides used in this study are listed in the Key resources table. Gateway cloning was performed according to the manufacturer's instructions (Invitrogen). All the constructs were verified by DNA sequencing (GATC-Biotech).

### Nematode strains and RNAi

*C. elegans* strains were cultured and maintained using standard procedures (*Brenner, 1974*). NPP-22 N-terminally tagged with mCherry using CRIPR/Cas9 was generated by SunyBiotech (https://www.sunybiotech.com/). The strain expressing LMN-1 N-terminally tagged with mCherry was constructed by mos1-mediated single-copy insertion (mosSCI) (*Frøkjaer-Jensen et al., 2008*). The engineered *lmn-1* gene contains a reencoded region in exon 4 essentially as described (*Penfield et al., 2018*).

RNAi was performed by the feeding method using HT115 bacteria essentially as described (*Kamath et al., 2001*), except that 2 mM of IPTG was added to the NGM plates and in the bacterial culture just prior seeding the bacteria. As a control, animals were exposed to HT115 bacteria harboring the empty feeding vector L4440 (mock RNAi). RNAi clones were obtained from the Arhinger library (Open Source BioScience) or were constructed.

### Feeding RNAi was performed as follows

Mild *gpr-1/2(RNAi)* was obtained by feeding L4 animals 14–16 hr at 20°C, whereas strong *gpr-1/2(RNAi)* was obtained by feeding L1 animals for 72 hr at 20°C before filming embryos.

For *efa-6* and *klp-7* inactivation, L4 animals were fed with bacteria at 15°C and embryos were filmed at 23°C.

For single *hcp-3* and *klp-7* inactivation or double *hcp-3*/*klp-7* inactivation, L4 larvae were fed for 14–16 hr at 15°C with bacteria producing *hcp-3* or *klp-7* dsRNA mixed volume to volume with control bacteria, or with bacteria producing *hcp-3* and *klp-7* dsRNA mixed volume to volume, respectively. The embryos were filmed at 23°C.

*plk-1*ts animals were fed with bacteria producing *lmn-1* dsRNA at 15°C from the L1 stage and briefly shifted at 25°C for 30 min before filming the embryos.

## Microscopy

For the analysis of the paired nuclei phenotype in live specimens by differential interference contrast (DIC) microscopy, embryos were obtained by cutting open young adult hermaphrodites using two 21-gauge needles. Embryos were handled individually and mounted on a coverslip in 3 µl of M9 buffer. The coverslip was placed on a 3% agarose pad. DIC images were acquired by an Axiocam Hamamatsu ICc 1 camera (Hamamatsu Photonics, Bridgewater, NJ) mounted on a Zeiss AxioImager A1 microscope equipped with a Plan Neofluar 100×/1.3 NA objective (Carl Zeiss AG, Jena, Germany), and the acquisition system was controlled by Axiovision software (Carl Zeiss AG, Jena, Germany). Images were acquired at 10 s intervals.

Live imaging was performed at 23°C using a spinning disk confocal head (CSU-X1; Yokogawa Corporation of America) mounted on an Axio Observer.Z1 inverted microscope (Zeiss) equipped with 491 and 561 nm lasers (OXXIUS 488 nm 150 mW, OXXIUS Laser 561 nm 150 mW) and sCMOS PRIME 95 camera (Photometrics). Acquisition parameters were controlled by MetaMorph software (Molecular Devices). In all cases ,a 63× Plan-Apochromat 63×/1.4 Oil (Zeiss) lens was used. Images were acquired at 2 s or 10 s intervals. Captured images were processed using ImageJ and Photoshop.

## Quantification and statistical analysis

The fluorescence intensity of GFP::LMN-1 and GFP::LMN-1 8A over time was measured from a single central focal plane with the ImageJ software in control and RNAi conditions. The same rectangle was used around the area of interest in each time-lapse acquisition, after background subtraction, the 'multi-measure' plugin was used to display the average signal intensity per pixel in each frame (10 s interval). Anaphase onset was defined as time 0. To allow direct comparison between control and RNAi conditions, the average signal intensity of GFP::LMN-1 or GFP::LMN-1 8A at the NE 120 s before anaphase was arbitrarily defined as 1. The data points on the graphs are the mean of the normalized GFP intensity measurements in control and RNAi conditions for the same defined region of interest (ROI).

The pronuclei area was measured using ImageJ software by thresholding the ROI defined by mCherry::LMN-1 and measuring the total area at each time point.

The intercentrosomal distances and embryos lengths were measured using the IMARIS software by manual tracking of the position of the centrosomes and the poles of the embryos, over time. Then, we used the following equation to obtain the corresponding intercentrosomal distance and embryo length at each time point:

$$d(x, y) = \sqrt{(x_2 - x_1)^2 + (y_2 - y_1)^2}$$

where the first point (first centrosome or embryo pole) is represented by $(x_1, y_1)$, and the second point (second centrosome or embryo pole) is represented by $(x_2, y_2)$. The data obtained were used to graph the ratio between intercentrosomal distance and embryo length (%) to monitor mitotic spindle elongation.

The results are presented as means ± SEM. The data presented on the graph *Figure 4E* were compared by the Mann-Whitney test. All calculations were performed using GraphPad Prism version 6.00 for Mac OS X, GraphPad Software, La Jolla, CA, https://www.graphpad.com/.

## Acknowledgements

The authors thank P Moussounda for her help with media preparation. The authors thank X Baudin and J Dumont for microscopy data acquisition and V Contremoulins for image analysis. The authors thank Orna Cohen-fix for stimulating discussions. The authors thank R Karess for critical reading of the manuscript and the anonymous reviewers for constructive comments. The authors acknowledge

the ImagoSeine core facility of the Institut Jacques Monod, member of IBiSA and France-BioImaging (ANR-10-INBS-04) infrastructures and the Institut Jacques Monod 'Structural and Functional proteomic platform.' GVA was supported by the Labex 'Who am I?' Laboratory of Excellence no. ANR-11-LABX-0071, the French Government through its Investments for the Future program operated by the French National Research Agency (ANR) under Grant no. ANR-11-IDEX-0005-01 and by the CONACYT Grant CVU 364106 and the SECTEI/162/2021 fellowship. NJ is supported by a funding 'Dynamic Research' from ANR-18-IDEX-0001, IdEx Université de Paris. Work in the laboratory of LP is supported by grants from 'Agence Nationale pour la Recherche' (ANR, France - ANR-17-CE13-0011) and by the 'Ligue Nationale Contre le Cancer' (Equipe labéllisée, France).

## Additional information

### Funding

| Funder | Grant reference number | Author |
|---|---|---|
| Labex | ANR-11-LABX-0071 | Griselda Velez-Aguilera |
| Agence Nationale de la Recherche | ANR-11-IDEX-0005-01 | Griselda Velez-Aguilera |
| Consejo Nacional de Ciencia y Tecnología | CVU 364106 | Griselda Velez-Aguilera |
| Secretaría de Educación Superior, Ciencia, Tecnología e Innovación | SECTEI/162/2021 fellowship | Griselda Velez-Aguilera |
| Agence Nationale de la Recherche | ANR-18-IDEX-0001 | Nicolas Joly |
| Université de Paris | IdEx | Nicolas Joly |
| Agence Nationale de la Recherche | ANR-17-CE13-0011 | Lionel Pintard |
| Ligue Contre le Cancer | Programme équipe labellisée | Lionel Pintard |

The funders had no role in study design, data collection and interpretation, or the decision to submit the work for publication.

### Author contributions

Griselda Velez-Aguilera, Conceptualization, Formal analysis, Funding acquisition, Investigation, Methodology, Validation, Writing - original draft, Writing – review and editing; Batool Ossareh-Nazari, Conceptualization, Investigation, Methodology, Writing – review and editing; Lucie Van Hove, Conceptualization, Methodology, Resources; Nicolas Joly, Investigation, Methodology, Resources; Lionel Pintard, Conceptualization, Formal analysis, Funding acquisition, Investigation, Methodology, Project administration, Resources, Supervision, Validation, Visualization, Writing - original draft, Writing – review and editing

### Author ORCIDs

Griselda Velez-Aguilera (iD) http://orcid.org/0000-0002-9662-8833
Lionel Pintard (iD) http://orcid.org/0000-0003-0286-4630

### Decision letter and Author response

Decision letter https://doi.org/10.7554/eLife.75382.sa1
Author response https://doi.org/10.7554/eLife.75382.sa2

## Additional files

### Supplementary files
• Transparent reporting form

### Data availability
All the raw data are provided in the manuscript.

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
