## [Editor Report]

This manuscript shows how the mitotic spindle helps to break apart the nuclear envelopes surrounding the maternal and paternal genomes so that they can be mixed together after fertilisation. This study will be interesting for cell biologists and biophysicists studying nuclear organization and mechanics. The work provides new insights into how pulling forces from the cell cortex influence the dynamics of nuclear rupture during mitosis.

---

## [Decision Letter]

**Decision letter after peer review:**

Thank you for submitting your article "Cortical microtubule pulling forces contribute to the union of the parental genomes in the *C. elegans* zygote" for consideration by *eLife*. Your article has been reviewed by 2 peer reviewers, and the evaluation has been overseen by a Reviewing Editor and Anna Akhmanova as the Senior Editor. The reviewers have opted to remain anonymous.

Essential revisions:

1. In the Introduction, the authors need to explain their focus on scission near the chromosomes. As written, the introduction describes lamina breakdown at the poles (presumably promoted by high levels of PLK-1) and at the site of scission and never explains why the authors focus their analysis on the scission site.

2. In the Introduction, the authors state on page 3, lines 64-65 that "The breakdown of the juxtaposed pronuclear envelopes of the oocyte and sperm begins with the formation of a membrane scission event". But the authors earlier point out that the breakdown near centrosomes is the initial event. This was somewhat confusing, although it became clear that the authors are focusing on the second event, which they call scission. The authors should clearly indicate that scission refers only to this second type of dis-assembly and not the first.

3. In reference to Figure 1, the authors refers to microtubules "pushing or pulling" on the matrix to promote breakdown, but the image shows a protrusion of the matrix that does not seem consistent with pushing. Do the authors typically protrusions, or a variety of deformations? More detail here would be helpful.

4. In Figure 1, the authors introduce the dis-assembly in control zygotes. The authors should explain if the dis-assembly seen near centrosomes is only visible using the LMN-1 marker and not the NPP-22 marker, and why the scission event is scene only with the NPP-22 marker and not the LMN-1 marker. Indeed, LMN-1 signal remains strong in a bright focus in the middle before the boundary disappears entirely. More information about why different markers are needed to score the two events would be helpful (and again the authors can better explain why they focus on scission and not the earlier step near centrosomes).

5. On page 7, lines 136-137, the authors state the spindle microtubules "dictate" the late triangular shape of the pronuclei, but this is an assumption and the authors do not provide any evidence that microtubules in fact are required for this shape.

6. In Figure 3, the authors need to provide more detail on how they quantified the fluorescence intensity. Did they use single focal planes, or maximum projects, or what? This information is not provided in the figure legend or methods. This information should be provided for all quantifications.

7. The authors should make it clear when introducing scission and discussing their results that the exact site of scission is dictated by the chromosomes. This is not mentioned when scission is first described, and it is not clear why scission would occur at such a specific site if stretching by elongation were the only other factor besides PLK-1 activity. The authors should also make it clear if their analysis provides the first evidence that the site of scission appears to be dictated by the organization of chromosomes to the metaphase plate.

8. On page 15, lines 339-340, the authors state that lmn-1 knockdown restored scission without affecting the levels and localization of LEM-2 or EMR-1. But in fact the distributions of both do appear very different in the control and lmn-1(RNAi) images. ERM-1 looks more like LEM-2 after lmn-1 knockdown, with much more ovlap, and both are much more rounded in the controls, with lmn-1 knockdown leading to a more triangular appearance. While this may not change their interpretation, the authors should acknowledge and try to explain these differences.

9. It is unclear whether nuclear membrane scission is a phenomenon controlled by mechanical forces, cell cycle, or both. Some of the experiments in the manuscript suggest mechanical forces control it, but some are not very supportive of this. For example, gpr-1/2 (RNAi) in lmn-1 8A supports the hypothesis that mechanical forces are important for nuclear scission. However, the lack of scission in hcp-3 (RNAi) in lmn-1 WT, where chromosome alignment is prevented, suggests a cell-cycle dependence mechanism. Can the authors elaborate more on this issue?

10. The relation between membrane scission and paired nuclei phenotype is unclear. It seems that paired nuclei phenotype in daughter cells is a direct consequence of lack of membrane scission earlier during first mitosis. The authors showed that in gpr-1/2 (RNAi) in lmn-1 WT, 83% of embryos have no nuclear membrane scission. However, only 5% of gpr-1/2 (RNAi) in lmn-1 WT embryos showed paired nuclei phenotype (Figure 2A). Is this an inconsistency? If not, could the authors explain this in more detail?

11. Similarly, why in klp-7 (RNAi) lmn-1 8A there is paired nuclei phenotype while the membrane scission happens (figure 5D), and in hcp-3 (RNAi) in lmn-1 WT, there is no paired nuclei phenotype, while membrane scission is completely prevented? It seems that one needs more knowledge/explanation about nuclear membrane scission and paired nuclei phenotype to understand these results. The short speculation at the top of page 11 does not seem enough.

12. The authors suggested an interesting mechanism for spindle length regulation in a one-cell stage embryo as a force balance between cortical pulling forces and membrane tension. To my knowledge, the role of the nuclear membrane had not been discussed for spindle length regulation. This could be interesting to highlight in conclusions.

13. The connection between plk-1ts experiments and the cortical pulling forces is not clear. The authors start with the proposal of testing the effect of pulling forces on membrane scission, but by this point in the manuscript, it seems that pulling forces have a secondary effect on scission. By the end, the authors argue that lamina depolymerization, chromosome alignment, and cortical pulling forces are all important for nuclear membrane scission. If so, which one is the primary factor? How are the others comparable to the primary factor?

---

## [Author Response]

Essential revisions:1. In the Introduction, the authors need to explain their focus on scission near the chromosomes. As written, the introduction describes lamina breakdown at the poles (presumably promoted by high levels of PLK-1) and at the site of scission and never explains why the authors focus their analysis on the scission site.

We thank the reviewers for the suggestion, and we apologize for the confusion. We did not provide enough details on the composition of the nuclear envelope and the different steps of its disassembly in the *C. elegans* zygote. We have extensively revised the introduction to clarify this point. In particular, we now better introduce the different steps of NEBD in the zygotes. We also now explain why we are focusing on the mechanisms regulating the site of membrane scission.

2. In the Introduction, the authors state on page 3, lines 64-65 that "The breakdown of the juxtaposed pronuclear envelopes of the oocyte and sperm begins with the formation of a membrane scission event". But the authors earlier point out that the breakdown near centrosomes is the initial event. This was somewhat confusing, although it became clear that the authors are focusing on the second event, which they call scission. The authors should clearly indicate that scission refers only to this second type of dis-assembly and not the first.

We have corrected the text to make it clear that pronuclear envelope breakdown, in particular, NPC disassembly and lamina depolymerization, starts in the vicinity of the centrosomes and then later occurs between parental chromosomes. Formation of membrane scission is another event of NEBD in the *C. elegans* zygote that occurs when the parental chromosomes mingle on the metaphase plate.

3. In reference to Figure 1, the authors refers to microtubules "pushing or pulling" on the matrix to promote breakdown, but the image shows a protrusion of the matrix that does not seem consistent with pushing. Do the authors typically protrusions, or a variety of deformations? More detail here would be helpful.

We typically observe protrusions of the nuclear envelope close to the centrosomes. The observation of protrusions is indeed more consistent with pulling than pushing. We have corrected the text accordingly.

4. In Figure 1, the authors introduce the dis-assembly in control zygotes. The authors should explain if the dis-assembly seen near centrosomes is only visible using the LMN-1 marker and not the NPP-22 marker, and why the scission event is scene only with the NPP-22 marker and not the LMN-1 marker. Indeed, LMN-1 signal remains strong in a bright focus in the middle before the boundary disappears entirely. More information about why different markers are needed to score the two events would be helpful (and again the authors can better explain why they focus on scission and not the earlier step near centrosomes).

We have now addressed this issue by mentioning more clearly in the introduction that lamina depolymerization and membrane scission are two separate events. For these reasons, we used different markers to visualize these two events. NPP-22 is a transmembrane nucleoporin that decorates the membranes during mitosis and behaves essentially as inner nuclear membrane proteins (e.g LEM-2).

5. On page 7, lines 136-137, the authors state the spindle microtubules "dictate" the late triangular shape of the pronuclei, but this is an assumption and the authors do not provide any evidence that microtubules in fact are required for this shape.

The reviewers are correct, we indeed do not prove that microtubules are required for this shape and we have corrected the text as follows:

“As more microtubules invaded the pronuclei space to capture paternal chromosomes, the overall surface of the pronuclei decreased (Figure 1E, Table 1 Source Data 1), and the pronuclei adopted a more triangular shape, possibly dictated by microtubules assembling the mitotic spindle in the space confined by the remnants of the pronuclear envelopes (Figure 1B) (Hayashi et al., 2012).”

6. In Figure 3, the authors need to provide more detail on how they quantified the fluorescence intensity. Did they use single focal planes, or maximum projects, or what? This information is not provided in the figure legend or methods. This information should be provided for all quantifications.

We thank the reviewers for bringing this important point. We have now included more details on the quantifications in the figure legends and in the methods.

“The fluorescence intensity of GFP::LMN-1 and GFP::LMN-1 8A over time was measured from a single central focal plane with the Image J software in control and RNAi conditions. The same rectangle was used around the area of interest in each time-lapse acquisition, after background subtraction, the “multi-measure” plugin was used to display the average signal intensity per pixel in each frame (10 s interval). Anaphase onset was defined as time 0. To allow direct comparison between control and RNAi conditions, the average signal intensity of GFP::LMN-1 or GFP::LMN-1 8A at the NE 120 s before anaphase was arbitrarily defined as 1.”

7. The authors should make it clear when introducing scission and discussing their results that the exact site of scission is dictated by the chromosomes. This is not mentioned when scission is first described, and it is not clear why scission would occur at such a specific site if stretching by elongation were the only other factor besides PLK-1 activity. The authors should also make it clear if their analysis provides the first evidence that the site of scission appears to be dictated by the organization of chromosomes to the metaphase plate.

Our analysis is not exactly the first evidence that the site of scission is dictated by the organization chromosomes. The Cohen-Fix lab previously showed that membrane scission is abrogated when chromosomes are not aligned as in *hcp-3(RNAi)* embryos (Rahman, et al., 2015). We have modified the introduction to make this point clear.

8. On page 15, lines 339-340, the authors state that lmn-1 knockdown restored scission without affecting the levels and localization of LEM-2 or EMR-1. But in fact the distributions of both do appear very different in the control and lmn-1(RNAi) images. ERM-1 looks more like LEM-2 after lmn-1 knockdown, with much more ovlap, and both are much more rounded in the controls, with lmn-1 knockdown leading to a more triangular appearance. While this may not change their interpretation, the authors should acknowledge and try to explain these differences.

The reviewers are correct; thanks for the suggestion. Here we only meant to say that *lmn-1* knockdown does not affect the recruitment of LEM-2 and EMR-1 to the nuclear envelope. However, it is clear that the localization/distribution of LEM-2 and EMR-1 is affected in *plk-1*ts embryos and restored upon partial *lmn-1* inactivation. We have modified the text to acknowledge and try to explain these differences. We have also included pics showing the distribution of GFP::LEM-2 and mCherry::EMR-1 in wild-type embryos in order to compare and discuss the difference in the distributions of these INM proteins in wild-type versus mutant conditions (see new Figure 6).

9. It is unclear whether nuclear membrane scission is a phenomenon controlled by mechanical forces, cell cycle, or both. Some of the experiments in the manuscript suggest mechanical forces control it, but some are not very supportive of this. For example, gpr-1/2 (RNAi) in lmn-1 8A supports the hypothesis that mechanical forces are important for nuclear scission. However, the lack of scission in hcp-3 (RNAi) in lmn-1 WT, where chromosome alignment is prevented, suggests a cell-cycle dependence mechanism. Can the authors elaborate more on this issue?

Pronuclear membrane scission is controlled both by mechanical forces and cell cycle mechanisms. This is supported by the fact that reduction of cortical microtubule traction forces [*gpr-1/2(RNAi)*] is sufficient to abrogate membrane scission (Figures 3B and 3C), even when the lamina is depolymerized and chromosomes perfectly align on the metaphase plate. Likewise, pronuclear membrane scission is prevented in absence of chromosome alignment [*hcp-3(RNAi)*].

10. The relation between membrane scission and paired nuclei phenotype is unclear. It seems that paired nuclei phenotype in daughter cells is a direct consequence of lack of membrane scission earlier during first mitosis. The authors showed that in gpr-1/2 (RNAi) in lmn-1 WT, 83% of embryos have no nuclear membrane scission. However, only 5% of gpr-1/2 (RNAi) in lmn-1 WT embryos showed paired nuclei phenotype (Figure 2A). Is this an inconsistency? If not, could the authors explain this in more detail?

While it was reported that a lack of membrane scission generally accompanies the paired nuclei phenotype, our data indicate this is not systematically the case. As the reviewers pointed out, most *gpr-1/2(RNAi)* embryos have no pronuclear membrane scission but still do not systematically present paired nuclei phenotypes, although the shape of the reforming nuclei is often abnormal in these embryos. How the membranes are remodeled during nuclear reformation to resolve this physical barrier is currently unclear. It should be noted that in most conditions reported in the literature resulting in the formation of embryos with a paired nuclei phenotype, lamina disassembly is either delayed or prevented. The persisting lamina by rigidifying the membrane constitutes a strong physical barrier that is not resolved during nuclei reformation and membrane expansion, resulting in paired nuclei phenotype.

11. Similarly, why in klp-7 (RNAi) lmn-1 8A there is paired nuclei phenotype while the membrane scission happens (figure 5D), and in hcp-3 (RNAi) in lmn-1 WT, there is no paired nuclei phenotype, while membrane scission is completely prevented? It seems that one needs more knowledge/explanation about nuclear membrane scission and paired nuclei phenotype to understand these results. The short speculation at the top of page 11 does not seem enough.

Given that cortical microtubules pulling forces are necessary for pronuclear membranes scission, and that increasing pulling forces induce a premature membranes scission event; we asked whether increased pulling forces [*klp-7(RNAi)*] could restore membrane gap in embryos, defective in lamina depolymerization (*lmn-1* 8A) or chromosome alignment [*(hcp-3(RNAi))*].

We observed that increasing pulling forces did not restore membrane scission in these conditions. Membranes scission did not occur in *klp-7(RNAi)*; *lmn-1 8A* embryos or in double *klp-7(RNAi), hcp-3(RNAi)* embryos.

These observations suggest that coordination between lamina depolymerization, chromosome alignment and cortical microtubules pulling forces is required to promote pronuclear membranes scission.

Please note that *hcp-3(RNAi)* embryos are impossible to score regarding the paired nuclei phenotype because chromosomes do not congress on the metaphase plates and fail to segregate in these embryos.

12. The authors suggested an interesting mechanism for spindle length regulation in a one-cell stage embryo as a force balance between cortical pulling forces and membrane tension. To my knowledge, the role of the nuclear membrane had not been discussed for spindle length regulation. This could be interesting to highlight in conclusions.

We thank the reviewers for the suggestion. We now highlight this connection between nuclear membrane remodeling and mitotic spindle elongation both in the abstract and in the conclusion section.

13. The connection between plk-1ts experiments and the cortical pulling forces is not clear. The authors start with the proposal of testing the effect of pulling forces on membrane scission, but by this point in the manuscript, it seems that pulling forces have a secondary effect on scission. By the end, the authors argue that lamina depolymerization, chromosome alignment, and cortical pulling forces are all important for nuclear membrane scission. If so, which one is the primary factor? How are the others comparable to the primary factor?

We do not think there is such a primary factor but rather a coordination between chromosome alignment, lamina depolymerization and mitotic spindle elongation. We have modified the concluding paragraph to clarify this.